# Oral Absorption of Middle-to-Large Molecules and Its Improvement, with a Focus on New Modality Drugs

**DOI:** 10.3390/pharmaceutics16010047

**Published:** 2023-12-28

**Authors:** Daigo Asano, Hideo Takakusa, Daisuke Nakai

**Affiliations:** Drug Metabolism and Pharmacokinetics Research Laboratories, Daiichi Sankyo Co., Ltd., 1-2-58, Hiromachi, Shinagawa-ku, Tokyo 140-8710, Japan; takakusa.hideo.yb@daiichisankyo.co.jp (H.T.); nakai.daisuke.jf@daiichisankyo.co.jp (D.N.)

**Keywords:** middle-to-large molecule, absorption enhancer, new modality, Lipinski’s rule of five, cyclic peptide, antisense oligonucleotide, target protein degrader, SNAC, C10, C8

## Abstract

To meet unmet medical needs, middle-to-large molecules, including peptides and oligonucleotides, have emerged as new therapeutic modalities. Owing to their middle-to-large molecular sizes, middle-to-large molecules are not suitable for oral absorption, but there are high expectations around orally bioavailable macromolecular drugs, since oral administration is the most convenient dosing route. Therefore, extensive efforts have been made to create bioavailable middle-to-large molecules or develop absorption enhancement technology, from which some successes have recently been reported. For example, Rybelsus^®^ tablets and Mycapssa^®^ capsules, both of which contain absorption enhancers, were approved as oral medications for type 2 diabetes and acromegaly, respectively. The oral administration of Rybelsus and Mycapssa exposes their pharmacologically active peptides with molecular weights greater than 1000, namely, semaglutide and octreotide, respectively, into systemic circulation. Although these two medications represent major achievements in the development of orally absorbable peptide formulations, the oral bioavailability of peptides after taking Rybelsus and Mycapssa is still only around 1%. In this article, we review the approaches and recent advances of orally bioavailable middle-to-large molecules and discuss challenges for improving their oral absorption.

## 1. Introduction

For many years, the pharmaceutical industry has primarily focused on the development of traditional small-molecule drugs (molecular weight (MW) ≤ 500). However, in recent times, there has been growing interest in new drug scaffolds such as antibody drug conjugates (ADCs), macrocycles, cyclic peptides, target protein degraders (TPDs), antisense oligonucleotides (ASOs), and small interfering RNA (siRNA). These novel drug modalities offer new therapeutic approaches that were previously unattainable with the existing modalities [1,2,3]. These emerging modalities are collectively referred to as new modality drugs [1,2,3,4,5,6], and they have beneficial features lacking in traditional small-molecule drugs. For example, macrocycles and cyclic peptides can bind to groove or cleft structures of target proteins with a large surface area, mimicking protein–protein interactions, while small molecules mainly interact with pocket structures of proteins [7,8,9,10]. Another advantage of macrocycles and cyclic peptides is the accessibility of intracellular targets, to which naked antibodies cannot bind. It is important to select an appropriate modality when we develop new drugs capable of addressing unmet medical needs.

Most new drug modalities are within the category of middle-to-large molecules, and thus, they tend to lack oral bioavailability, which is one of the greatest advantages of small-molecule drugs. Considering that oral administration is superior to other dosing routes (e.g., intravenous and subcutaneous) given its noninvasiveness and safety [11], parenteral administration of new drug modalities limits their potential use by patients. The reason for the poor oral bioavailability of new drug modalities can be explained by Lipinski’s rule of five [12] defining the necessary properties for oral absorption as follows: molecular weight (MW) ≤ 500, number of hydrogen bond donors (HBDs) ≤ 5, number of hydrogen bond acceptors (HBAs) ≤ 10, and octanol-water partition coefficient (LogP) ≤ 5. Veber et al. suggested additional rules for orally absorbed drugs [13], namely, that the number of rotatable bonds should be fewer than 10 and the topological polar surface area (TPSA) should be less than 140 Å^2^. As illustrated by octreotide, an example cyclic peptide, the physicochemical properties of most of the new drug modalities exceed the thresholds in Lipinski’s rule of five and these additional rules [14] (Figure 1). More specifically, most middle-to-large molecules tend to lack membrane permeability, which is essential for gastrointestinal absorption after oral dosing.

This background indicates the need for technical innovation in order to develop middle-to-large molecules with oral bioavailability. To date, various different technologies have been applied to the development of orally bioavailable peptides, but most of these attempts failed until the emergence of two game-changers: Rybelsus^®^ tablets and Mycapssa^®^ capsules [15,16]. Rybelsus tablets and Mycapssa capsules contain salcaprozate sodium (SNAC) and sodium caprylate (C8), both of which are absorption enhancers, thereby enabling the gastrointestinal absorption of semaglutide [17,18] (a GLP-1 agonistic peptide with a molecular weight of approximately 4100) and octreotide [19] (a somatostatin receptor agonistic peptide with a molecular weight of 1019), respectively. Patients can orally take Rybelsus tablets or Mycapssa capsules to treat type 2 diabetes or acromegaly at home. To commemorate this new era of bioavailable peptide drugs, this review article summarizes the technological progress in improving the oral absorption of new drug modalities and discusses future prospects to improve absorption enhancement. Since numerous review articles about oral absorption enhancement technology have already been published [20,21,22,23,24,25,26,27,28,29,30,31,32,33,34,35], this manuscript provides a brief explanation of these technologies in Section 2. However, the main focus of this manuscript involves shedding light on new aspects of orally bioavailable middle-to-large molecules, mainly based on information in the recent (from 2020 to 2023) literature available in PubMed, in order to keep readers up to date with the latest technology.

## 2. Technology to Improve Oral Absorption of Middle-to-Large Molecules

There are two strategies to achieve the oral absorption of middle-to-large molecules: chemical modification and the utilization of an absorption enhancer. Each approach has its own advantages and limitations, as summarized in Table 1.

### 2.1. Chemical Modification to Acquire Membrane Permeability and Chameleonic Property

The oral bioavailability of peptides is generally less than 1% in animal models. An exception to this is the peptide cyclosporin, which is a natural product isolated from fungi discovered in 1971 [41]. Despite the large molecular weight of 1202, cyclosporin shows a good bioavailability of approximately 30% [41] due to novel formulations and its unique structure (Figure 2). In terms of the formulation, a self-microemulsifying drug delivery system (Neoral^®^) has already been developed to overcome the poor solubility of cyclosporin [39]. Its cyclic peptide structure with *N*-methylated amide bonds and unnatural amino acids confers it with good metabolic stability against peptidases and “chameleonic property” [37]. This latter property means that the conformation of cyclosporin dramatically changes upon transition from aqueous to lipophilic conditions, achieving good membrane permeability and oral bioavailability (Figure 3) [42,43]. The reason for such a change in structure in a manner depending on the conditions can be explained by the change in the mode of hydrogen bonding, from interaction with water molecules in the aqueous environment to intramolecular hydrogen bonding in the lipid membrane. Extensive efforts are now being dedicated to synthesizing artificial cyclic peptides or middle-to-large molecules with membrane permeability using various evaluation methods, such as the measurement of lipophilic permeability efficiency (LPE) [44], experimental polar surface area (EPSA) [45,46], nuclear magnetic resonance (NMR) [47,48], and X-ray analysis [36] (Table 2).

### 2.2. Utilization of Absorption Enhancers

While the maximum molecular weight of orally bioavailable membrane-permeable peptides is around 1200 [38], absorption enhancers can improve the oral absorption of middle-to-large molecules with molecular weights more than 1000 (e.g., the molecular weight of semaglutide contained in Rybelsus is approximately 4100). There are many kinds of absorption enhancers, such as cell-penetrating peptides (CPPs) (i.e., TAT or octa-arginin) [54], claudin modulators [55], ethylenediaminetetraacetic acid (EDTA) [56], and bile acid [57] (Table 3). As shown in Table 3, most of the absorption enhancers are considered safe additives, although no information is available on the toxicity of some of them. Recently, Otsuki et al. discovered cyclic DNP peptide [58,59], which could enhance the intestinal absorption of insulin. The majority of CPPs, such as TAT and octa-arginin, are cationic peptides containing multiple lysine and arginine moieties, and their target molecules are heparan sulfate proteoglycans [60]. However, cyclic DNP peptide does not contain basic amino acids and likely interacts with integrin α_v_β_3_ [58]. With regard to clinical application, two of the most well-known absorption enhancers are fatty acids (C8 and C10) and SNAC [15,16,25]. These absorption enhancers are defined as generally recognized as safe (GRAS) substances by the FDA. Despite their similar chemical structures (Figure 4), they have different origins and modes of action, as described in the following sections.

#### 2.2.1. Fatty Acids (C8 and C10)

Origin: In the 1980s, fatty acid emulsions were known to have an absorption-enhancing effect. Van Hoogdalem et al. thought that medium-chain fatty acids contained in emulsions might be nontoxic and effective for poorly absorbed drugs, and they demonstrated that the rectal absorption of cefoxitin could be improved by the concomitant administration of caprylic acid (C8) and capric acid (C10) [88]. C8 and C10 are originally derived from food ingredients, and these additives are abundant in dairy milk products [89]. Additionally, C8 and C10 have been clinically applied for the development of many drugs in humans, as follows.
Epocelin^®^ suppositories (antibiotics prescribed in Japan [90]) contain C10 to enhance the rectal absorption of ceftizoxime sodium in humans.Krug et al. reported that C10 improved the rectal absorption of ampicillin in humans [91].Tuvia et al. reported that C8 enhanced the oral absorption of octreotide in humans [92].Halberg et al. and Tillman et al. reported that C10 enhanced the oral absorption of insulin [40] and antisense oligonucleotides [93] in humans, respectively.

Mechanism: The most widely accepted mechanisms behind the absorption-enhancing effects of fatty acids include the opening of tight junctions of cellular membranes by the activation of phospholipase C, increased calcium level, and altered localization of tight junction proteins, because the transepithelial electrical resistance (TEER) value of cells was found to be significantly decreased by the addition of fatty acids (C8, C10, etc.) [65,94,95,96]. Another potential mechanism behind the absorption enhancement by C8 and C10 is the perturbation of cellular membranes [97,98]. Nonetheless, the entire absorption-enhancing mechanism by fatty acids remains unclear.

Safety: Fatty acids have been regarded as safe additives [89]. Moreover, previous in vivo studies indicated that C8 and C10 are safe additives, as follows.
Leonard et al. reported that the oral administration of C10 at 1000 mg/kg for 7 days did not cause any side effects in dogs [99].Raoof et al. reported that the oral administration of C10 at 990 mg/body (as three ISIS104838-containing tablets) for 7 days was safe in dogs [100].Tuvia et al. reported that the oral administration of C8 (as octreotide-containing oily suspension) for 9 months was tolerated with minor toxicity in monkeys [96].Halberg et al. reported that the oral administration of C10 at 550 mg/body (as I338 tablets) for 8 weeks was well tolerated in humans [40].

#### 2.2.2. SNAC

Origin: SNAC was discovered by Emisphere Technology (now Novo Nordisk) in the 1990s. Emisphere investigated drug-loaded proteinoid microspheres composed of thermally condensed amino acids for oral medication [101]. Then, they derivatized the α-amino acids by *N*-acylation to enhance the oral absorption of proteins [102]. After testing numerous non-α-amino acid derivatives [103], they found that SNAC was one of the most effective absorption enhancers [104]. Intramolecular hydrogen bonding between phenolic hydrogen and a carbonyl moiety adjacent to the phenol ring is essential for the planar structure and absorption-enhancing effect of SNAC-related compounds, as interpreted from their chemical structure [103]. Therefore, although C10 and SNAC appear to have similar structures (Figure 4), they were discovered independently.

Mechanism: The entire mechanism behind the absorption enhancement by SNAC has not been fully elucidated, nor has that for fatty acids (C8 and C10). Some reports demonstrated that the TEER value of cells was decreased by the addition of SNAC [104,105], suggesting enhancement of the paracellular pathway. Meanwhile, there are other reports suggesting that SNAC could improve transcellular permeability without a significant decrease in the TEER value or the occurrence of cell damage [61,106,107]. A recent paper supports the latter mechanism. In this paper, apparent permeability across NCI-N87 cells was significantly enhanced by SNAC and EDTA, a paracellular enhancer, but an increased intracellular concentration of semaglutide was observed only with SNAC but not with EDTA [108]. Therefore, SNAC is considered to be an enhancer of transcellular permeability via the complex formation of transported compounds with SNAC [106] or inhibition of digestive enzymes [108]. Buckley et al. reported the very interesting experimental results that the gastric absorption of semaglutide was enhanced by SNAC when Rybelsus tablets were administered to pylorus-ligated dogs [108]. Because absorption enhancers have been used to improve intestinal or colorectal absorption for a long time, this evidence overturned the conventional wisdom regarding absorption enhancers. At the same time, several questions have arisen in this context:Is SNAC ineffective against intestinal permeation?Are other permeation enhancers (e.g., C8 and C10) effective against gastric permeation?Why do SNAC and other permeation enhancers have different sites of action, the stomach and the intestine, respectively?

To answer these questions, more detailed research on the mode of action is expected in the future.

Safety: Riley et al. reported that the no-observed-adverse-effect level (NOAEL) was 1000 mg/kg in a 13-week repeated-dose toxicity study of SNAC in male and female Wistar rats [109]. Recently, labeling materials of Rybelsus approved by FDA have already been disclosed. Based on this document [64], Novo Nordisk conducted various kinds of safety assessments of SNAC, in which the NOAELs were 500 and 500 mg/kg/day in male and female mice (13-week toxicity study), 500 and 75 mg/kg/day in male and female rats (104-week toxicity study), and 300 and 300 mg/kg/day in male and female monkeys (39-week toxicity study), respectively. The tolerability of SNAC in humans has been confirmed in various clinical studies and marketed medication (Eligen B12 and Rybelsus containing 100 and 300 mg of SNAC, respectively) [18,110].

### 2.3. Utilization of Special Formulations with an Absorption-Enhancing Effect

As shown in Table 4, most of the formulations for middle-to-large molecules or biomolecules containing absorption enhancers (i.e., C10) are enteric-coated to avoid degradation by acidic pH and digestive enzymes in the gastrointestinal tract. Notable exceptions are Rybelsus and EligenB12 tablets. Because SNAC improves gastric permeation, it is designed as immediate-release tablets. Although semaglutide is generally unstable in the presence of pepsin, SNAC released from Rybelsus could neutralize acidic pH in the stomach and decrease the hydrolytic activity of pepsin. Since solid formulations are essential for the clinical application of drugs, various kinds of special formulations are currently under evaluation (Table 4).

## 3. Recent Challenges of Orally Bioavailable Middle-to-Large Molecules

### 3.1. Application of Chemical Modification

#### 3.1.1. Cyclic Peptides

To date, numerous attempts have been made to synthesize orally bioavailable cyclic peptides like cyclosporin [120]. Among them, one of the largest cyclic peptides with high bioavailability is the cyclic decapeptide synthesized by Novartis [36]. The intramolecular hydrogen bonding and good membrane permeability of cyclic decapeptides were demonstrated using NMR and X-ray analyses, as well as in vitro experiments [36]. Although the best cyclic decapeptide (compound **9**, MW: 987 in [36]) showed approximately 100% BA in rats, oral BA (1% to 46%) of other cyclic decapeptides did not necessarily correspond with membrane permeability, suggesting that systemic exposure of these peptides is determined not only by permeability but also by other pharmacokinetic and/or physicochemical factors.

To elucidate the factors determining the oral BA of cyclic peptides, we evaluated the oral absorption of cyclic decapeptide A (Figure 5, MW: 1091) in detail [121]. Cyclic decapeptide A showed good membrane permeability in vitro (3.1 × 10^−6^ cm/s in MDCK cells) and solubility (640 μg/mL in JP2 solution), but its oral bioavailability in mice was less than 1% [121]. When cyclic decapeptide A was administered to mice pretreated with GF120918 (P-glycoprotein (P-gp) inhibitor, also known as elacridar) and 1-aminobenzotriazole (ABT) (cytochrome P450 (P450) inhibitor), the oral bioavailability approached 100% (Figure 6), suggesting that its oral absorption is largely inhibited by P-gp efflux and P450 metabolism [121]. It is worth noting that both P-gp and P450 are important factors for the oral absorption of cyclic peptides, as well as small-molecule drugs [122,123]. In general, middle-to-large molecules are easily recognized by P-gp [124,125]. Therefore, orally bioavailable peptides need not only membrane permeability but also the avoidance of a high affinity toward P-gp. We also found that cyclic decapeptide A did not undergo hydrolytic metabolism by pepsin, pancreatin, plasma, liver, and intestinal homogenates from mice, but it was mainly metabolized by P450 in vitro [121]. A metabolite identification study indicated that cyclic decapeptide A was converted into a de-ethylated metabolite in the liver and intestinal microsome fractions from mice (Figure 5 and Figure 7) [121]. *N*-Alkylation is suggested to be one of the chemical modification methods to obtain orally bioavailable peptides [126,127]; however, based on our results, the alkyl moiety can easily be recognized by P450. Thus, metabolic stability against P450 should be taken into consideration when *N*-alkylation is applied to peptides. Based on these experimental results, the reason for the poor bioavailability of cyclic decapeptide A was found to be extensive P-gp-mediated efflux and P450-mediated rapid metabolism into de-ethylated metabolites (Figure 8) [121]. Since a similar synergic elimination of small-molecule drugs by P-gp and P450 was also reported [122,123], it is noted that escape from these enzymes is important for not only small-molecule drugs but also cyclic peptides.

Although there is no approved cyclic peptide with oral bioavailability (more than 10%) and systemic efficacy after the discovery of cyclosporin, multiple pharmaceutical companies such as Chugai Pharmaceutical Co., Ltd. (Roche) [126,127], Shionogi Pharma Co., Ltd. [128], Merck & Co, Inc. [129,130], and PeptiDream Inc. (PeptiAID Inc.) [131,132] have been focusing on the development of orally bioavailable peptides. In fact, some of their peptides have already been tested in clinical studies. Merck has already disclosed that MK-0616 (MW: 1616) showed good potential and oral exposure in humans [129,130].

#### 3.1.2. TPD

Target protein degrader (TPD) is a heterobifunctional molecule that connects ligands for E3 ligase (e.g., von Hippel–Lindau tumor suppressor, cereblon, apoptosis proteins, and mouse double minute 2 homolog) and those for target proteins (e.g., androgen receptor, estrogen receptor, B-cell lymphoma-extra-large, bromodomain-containing protein 9, Bruton’s tyrosine kinase, epidermal growth factor receptor, interleukin-1 receptor-associated kinase 4, and signal transducer and activator of transcription 3) [133,134,135]. Recently, several TPDs (ARV-110 (MW: 812), ARV-471 (MW: 724), ARV-766 (MW: 808), DT2216 (MW: 1542), FHD-609 (MW: 829), NX-2127 (MW: 720), NX-5948 (MW: 807), etc.) have entered clinical trials [136]. Their chemical structure depends on the combination of the binders to target proteins, E3 ligase ligands (e.g., VH032, AZ-A, AZ-B, thalidomide, pomalidomide, lenalidomide, methylbestatin, LCL161 derivative, and nutlin-3) and the linkers (e.g., PEG, alkyl, glycol, alkyne, triazole, piperazine, and piperidine) [134]. The chemical structures of representative TPDs, ARV-110 and ARV-471, are shown in Figure 9. Despite the middle-to-large molecular weights of TPDs ranging from 600 to 1600 and their physiochemical properties being outside the rule of five, many TPDs, including ARV-110 and ARV-471, are under development as oral medications [136]. Since TPDs tend to have an affinity for P-gp [137,138], the avoidance of extensive P-gp recognition would be a key factor for orally bioavailable TPDs in humans, similar to cyclic peptides. The clinical outcomes of ongoing TPDs that are currently under evaluation would provide us with further information about the ADME characteristics of TPDs. Arvinas Inc. has already announced good oral exposure to ARV-110 and ARV-471 in humans [139,140,141] and suggested that the physicochemical parameters required for the oral absorption of TPDs appear to differ from the general criteria for rule of five drugs (e.g., MW ≤ 950, unsatisfied HBD ≤ 2, HBA ≤ 15, TPSA ≤ 200, number of rotatable bonds ≤ 14, cLogP ≤ 7, calculated octanol-water distribution coefficient (cLogD) ≤ 6, and number of aromatic rings (NAr) ≤ 5) [142].

#### 3.1.3. Other Middle-to-Large Molecules beyond the Rule of Five

The number of FDA-approved orally bioavailable middle-to-large molecules is increasing [143]. Most of them can be classified as diverse structures, including macrocycles [144,145], and they might also possess chameleonic property [37,146]. Examples of their structures are shown in Figure 10. The requirements for bioavailable middle-to-large molecules have been assessed by investigating their physicochemical properties. Doak et al. suggested both an “extended” rule of five (MW ≤ 700, HBD ≤ 5, TPSA ≤ 200 Å^2^, number of rotatable bonds ≤ 20, and 0 ≤ cLogP ≤ 7.5) and “limits” of rule of five (MW ≤ 1000, HBD ≤ 6, HBA ≤ 15, TPSA ≤ 250 Å^2^, number of rotatable bonds ≤ 20, and −2 ≤ cLogP ≤ 10) [143]. In addition, DeGoey et al. demonstrated a similar observation that middle-to-large molecules with MW ≤ 1132, TPSA ≤ 229 Å^2^, and −5.5 ≤ cLogP ≤ 13.3 could be bioavailable and reached the conclusion that the “AB-MPS” score calculated based on the following equation is a good indicator of oral bioavailability [147].
AB-MPS = Abs (cLogD − 3) + NAr + number of rotatable bonds

Middle-to-large molecules with AB-MPS less than 15 have a chance of being absorbed from the gastrointestinal tract [147].

### 3.2. Application of Absorption Enhancers and/or Special Formulations with an Absorption-Enhancing Effect

#### 3.2.1. Peptides

A representative clinical study for peptides was conducted after the oral administration of insulin formulated with C10 [40]. This study indicated that a blood glucose-lowering effect was observed in humans after the oral administration of a C10-containing formulation of insulin, suggesting that C10 can actually enhance the gastrointestinal permeability of insulin in humans [40]. However, this formulation has never been launched because of the high manufacturing cost [40]. Therefore, in the pharmaceutical industry, there is a need to consider the cost of manufacturing peptide formulations to provide high doses due to limited bioavailability. This seems to be the case with C8- and SNAC-containing formulations. Although Mycapssa capsules (with C8) and Rybelsus tablets (with SNAC) have been successfully approved and marketed, their oral bioavailability was only 0.7% in humans [19] and 1% in dogs [108] and humans [20], respectively.

To explore the room for improvement of absorption enhancer-containing formulations, we performed fundamental research on animals [148]. The first experiment involved the dose optimization of SNAC in male rats, where SNAC (10–1000 mg/kg) and daptomycin (10 mg/kg) (Figure 11), a cyclic peptide with low membrane permeability (0.3 × 10^−6^ cm/s in MDCK cells) and MW of 1621, were co-administered to male rats, and the plasma exposure level of daptomycin in male rats was measured [148]. The results showed that the plasma exposure of daptomycin in male rats increased with increasing the SNAC dose from 100 mg/kg to 1000 mg/kg (Figure 12) [148]. The effective SNAC dose of 100 to 1000 mg/kg in rats is consistent with previous reports indicating that the oral dose of absorption enhancers (SNAC-related compounds and C10) in animals generally ranged from 25 to 800 mg/kg [102,103,149,150,151,152]. One important question here is the difference in the required SNAC dose between rats (100 to 1000 mg/kg) and humans (300 mg in Rybelsus tablets). Novo Nordisk reported that the absorption-enhancing effect was saturated at a SNAC dose of 300 mg in humans, and a greater effect was not observed with 600 mg of SNAC [108]. Regarding the Mycapssa capsule, its C8 content has never been disclosed, but it is estimated to contain 100 mg of C8, assuming that octreotide (20 mg) and C8 account for 3% and 15% of the composition by weight, respectively, based on patent information [153]. It is generally noted that a liquid suspension or solution for oral administration can be prepared in animal experiments, while a solid formulation (tablet or capsule) is used in clinical settings. In the rat study, the dosing solution was prepared by dissolving daptomycin and SNAC in sodium bicarbonate buffer at pH 9, because SNAC is highly dissolved in alkaline buffer due to its acidity. Therefore, one possible explanation for the discrepancy in SNAC dose between animals and humans is the difference in dosing forms (suspension/solution or tablet/capsule). In other words, solid formulations might be able to minimize the required amount of SNAC by delivering SNAC to the appropriate region of the stomach in a more effective manner than solution administration. Another possible explanation for the smaller amount of SNAC in humans than in animals is the limitations of the current formulation technology. A very high dose of SNAC, such as more than 600 mg in the tablets, might be unable to exhibit an absorption-enhancing effect due to solubility or diffusion limitations in humans.

When daptomycin (5–10 mg/kg) and SNAC (200 mg/kg) were co-administered to monkeys and dogs, elevated plasma exposure to daptomycin was also observed in these animals (Figure 13), indicating the permeability-enhancing effect of SNAC across species [121]. Note that the PK experiments above were conducted in fasted animals, because the absorption-enhancing effect of SNAC can be strongly decreased by food intake [108]. According to the package insert of Rybelsus tablets, these tablets should be administered to patients before the first food [15].

As mentioned above, the absorption enhancement of peptides with SNAC can be easily evaluated in animals, and similar experiments have been performed with fatty acids, but the mode of interaction of peptides with SNAC and fatty acids has not been elucidated yet. To explore the peptide specificity, we performed a PK study of octreotide derivatives (octreotide (MW: 1019), lanreotide (MW: 1096), and pasireotide (MW: 1047)) (5 mg/kg) (Figure 14) with SNAC (200 mg/kg) in rats [148]. While SNAC’s effect of enhancing oral absorption was observed with octreotide and lanreotide, the plasma exposure of pasireotide was not increased by SNAC (Figure 15) [148]. A similar tendency was also described in the literature, where the oral absorption of liraglutide (Figure 16A, MW: ca. 3800), a GLP-1 analog like semaglutide (Figure 16B, MW: ca. 4100), was not enhanced by SNAC [108]. Thus, the absorption-enhancing effect of SNAC is sometimes ineffective for peptides analogous to bioavailable peptides upon the concomitant administration of SNAC. Additional research on the detailed structure–activity relationship between peptides and SNAC or fatty acids (C8 and C10) would be expected in the future to make better use of these absorption enhancers.

#### 3.2.2. Oligonucleotides

Oligonucleotide therapeutics have been attracting attention as a new treatment modality for a range of diseases that have been difficult to target by conventional approaches. As of 17 June 2023, oligonucleotide therapeutics have been approved, including 10 antisense oligonucleotides (ASOs) and 5 small interfering (si)RNAs for treating cardiovascular, neuromuscular, and central nervous system diseases [154,155,156,157]. One of the ADME-related characteristics common to oligonucleotide therapeutics is poor oral absorption, which is mainly due to low membrane permeability resulting from their molecular weight and hydrophilicity [158,159,160,161]. Therefore, orally administered oligonucleotide therapeutics have not yet been marketed, and intravenous (five drugs) or subcutaneous (seven drugs) administration has been adopted when systemic exposure is intended. Although the clinical application of oligonucleotide therapeutics has been achieved by IV- or SC-based systemic administration and local administration (e.g., IVT, IT, and IM), the oral delivery of oligonucleotides still holds potential clinical benefits and attractiveness because of its convenience, satisfactory medication compliance, and avoidance of injection site reactions, particularly when repeated administration is needed. In addition, in the case where the target is in the small intestine or liver, oral dosing could improve drug delivery to these organs through the first pass extraction effect. Therefore, various approaches such as the use of absorption enhancers and nanocarriers have been investigated to improve the oral absorption of ASOs and siRNAs.

Representative studies examining oral formulations of oligonucleotides with bioavailability data are summarized in Table 5. In the case of ASOs with a phosphorothioate backbone, poor intestinal permeability due to their charged and hydrophilic nature is the major hurdle to oral delivery, and thus, the formulation with C10, an absorption enhancer, has been intensively studied in both preclinical studies on animals and humans to improve oral bioavailability. Raoof et al. first evaluated the effect of this enhancer on the oral absorption of a 2′-O-methoxyethyl (2′-MOE)-modified phosphorothioate ASO, ISIS104838 (MW: ca. 7300), targeting human tumor necrosis factor alpha (TNF-α) mRNA in pigs [150]. Plasma concentrations of ISIS104838 after intrajejunal (IJ) administration at a dose of 10 mg/kg with C10 were measured by a HPLC/UV analysis, and the bioavailability relative to IV dosing at 2 mg/kg was calculated to be 1.7–2.8% by dose normalization. Then, a tablet formulation of ISIS104838 (80 mg) with C10 (330 mg) was tested in dogs, and the oral bioavailability after once-daily administration ranged from 1.1% to 1.7% relative to IV [100]. The bioavailability in major tissues was also evaluated in this study, and it was found to be dependent on tissue type, ranging from 2.0% to 4.3% relative to IV. The reason for the higher bioavailability in tissues than in plasma was considered to be the underestimation of the plasma concentration due to the limited sensitivity of the bioanalytical method used. The oral administration of ISIS104838 with C10 was further studied in humans, where 15 healthy subjects received four tablet formulations, changing the coating or drug to a C10 composition, in a crossover manner. The oral bioavailability of the tablet formulations was calculated relative to the dose-normalized historical parenteral plasma AUC after SC administration, ranging from 7.2% to 12.0%. This series of data on ISIS104838 suggested the possibility of practically applying orally administered ASOs by using absorption enhancers.

Gennemark et al. recently demonstrated the oral delivery of a highly potent ASO targeting PCSK9 mRNA, known as AZD8233 or ION-86366 (MW: ca. 6900), in which the chemical modification with constrained ethyl (cEt) chemistry and liver targeting by N-acetylgalactosamine (GalNAc) conjugation were applied to improve its potency [162]. To evaluate the oral delivery of AZD8233 with sodium caprate, a study of its single intrajejunal administration using jejunal-cannulated rats was performed, because the oral administration of tablets to rodents is not feasible. The liver concentrations of AZD8233 48 h after IJ and SC administrations at various doses were determined by a hybridization ELISA method, and the liver bioavailability of IJ dosing relative to SC was calculated to be 5.3%. Then, oral delivery of the tablet formulation was evaluated in a non-rodent study, where the concentrations of AZ8233 in the plasma, liver, and kidney were measured after repeated oral daily administration of a tablet containing 700 mg of sodium caprate and 3 or 20 mg of AZD8233 for 1 or 4 weeks. The result revealed liver bioavailability of 7.0–7.4%, which was about fivefold higher than the plasma bioavailability (1.3–1.8%), probably due to the active liver uptake by the GalNAc ligand and the first pass extraction effect. In addition, the bioavailability in the liver was significantly higher than that in the kidney (1.2–1.6%), suggesting the beneficial selectivity in tissue exposure between the liver and other organs with regards to the efficacy/safety margin. Based on these preclinical observations, liver exposure and PD parameters (PCSK9 knockdown and LDL cholesterol) in humans after oral administration were simulated, and it was suggested that a repeated oral daily dose of 15 mg/day would lead to PD marker changes comparable to those observed after SC administration at 25 mg/month.

As one of the other approaches for the oral delivery of ASOs, the formulation with a biodegradable albumin polymer matrix was reported to improve the oral absorption of an ASO targeting nuclear factor kappa B (NF-kB) mRNA [163]. Although significantly high oral bioavailability (70% relative to IV) was reported, further research appears to be necessary to demonstrate its mechanism and applicability.

For the oral administration of siRNAs, nanocarrier-based delivery technologies have been explored and tested in preclinical settings. The delivery system needs to overcome multiple physiological barriers, such as destabilization of the nanocarrier-siRNA complex in the harsh gastrointestinal environment, the electrostatic trapping of nanoparticles with a positively charged surface by the negatively charged components in the gastrointestinal mucus, and endosomal trapping in the target cells resulting in the insufficient release of siRNA in the cytosol.

One of the extensively studied biomaterials used in nanocarriers for the oral delivery of siRNA is chitosan, a biocompatible polysaccharide [164,165]. It can prolong the residence time on the epithelial surface and facilitate paracellular drug transport due to its mucoadhesive and mucopermeable nature. Ballarín-González et al. demonstrated by Northern blotting and quantitative PCR analysis that siRNA encapsulated in chitosan-based nanoparticles retained the structural integrity and was distributed in the stomach, small intestine, and colon after oral administration to mice [166]. In addition, Han et al. formulated chitosan-based nanocarriers loaded with fluorescence (TAMRA)-labeled siRNA and evaluated the exposure in plasma and tissues following oral administration to tumor-bearing mice [167]. The TAMRA-siRNA contents in the plasma and the supernatant of the tissues were determined by fluorescence measurement and calculated as the percentage of the total amount. The results revealed that the oral administration of TAMRA-siRNA by chitosan-based nanocarrier formulations was associated with significantly higher exposure in the plasma and tumor than oral administration of the naked siRNA. Although accurate concentration data or subsequent oral bioavailability were not determined in this study, approximately 6–7% of the total TAMRA-siRNA signals were found in the plasma at 4 and 12 h after the oral administration of a chitosan-containing formulation.

Recently, Wei et al. developed small, fluorinated nanocapsules for the efficient oral delivery of siRNA targeting tumor necrosis factor α (TNF-α) [168]. The nanocapsules are designed to be stable in the gut due to their shell structure with disulfide cross-linkages and are designed with a relatively small particle size (~30 nm) to facilitate diffusion in the mucus layer. The particle tracking assay demonstrated that the fluorinated nanocapsules were more able to diffuse than the control nanocapsules without fluorocarbon. In addition, an in vitro experiment using porcine mucin revealed that the degree of adsorption to mucin decreased in a fluorocarbon content-dependent manner, suggesting that the modification of fluorocarbon could facilitate the penetration of mucus by preventing adsorption to mucin glycoproteins. In a PK study, fluorinated nanocapsules loaded with TNF-α siRNA were orally and intravenously administered to mice, and plasma concentrations of siRNA were determined by a PCR-based method. The calculated oral bioavailability result for the best optimized formulation of fluorinated nanocapsules was 20.4% relative to IV injection.

**Table 5 pharmaceutics-16-00047-t005:** Representative studies for the oral delivery of oligonucleotide with bioavailability data.

Target Gene (Name of Oligonucleotide)	MW	Type of Oligonucleotide	Formulation/Modification for Oral Delivery	Species	Bioavailability	Bioanalytical Method	Reference
TNF-α (ISIS104838)	ca. 7300	PS-ASO 2′-MOE	C10	Pig	IJ relative to IV: 1.7–2.8% in plasma	HPLC/UV	[150]
				Dog	PO relative to IV: 1.1–1.7% in plasma 1.3–4.3% in tissues	HPLC/UV	[100]
				Human	PO relative to SC: 7.2–12.0% in plasma	hybridization ELISA	[93]
NF-kB	Unknown	ASO, modification unspecified	biodegradable albumin polymer matrix	Rat	PO relative to IV: 70% in plasma	OliGreen fluorescence assay	[163]
PCSK9 (AZD8233, ION-863633)	ca. 6900	PS-ASO GalNAc cET chemistry	C10	Rat	IJ relative to SC: 5.3% in liver	hybridization ELISA	[162]
				Dog	PO relative to SC: 1.3–1.8% in plasma 7.0–7.4% in liver 1.2–1.6% in kidney		
TNF-α	Unknown	siRNA	fluorinated nanocapsules	Mouse	PO relative to IV: 20.4% in plasma	PCR-based method	[168]

## 4. Conclusions and Future Perspectives

The emergence of Rybelsus tablets and Mycapssa capsules has ushered in a new era in which patients can orally take pharmacologically active peptide drugs with a molecular weight of 1000 to 4100. However, there are still many challenges to be overcome, such as poor bioavailability (approximately 1%) and the selection of pharmacologically active peptides with permeability that can be improved by absorption enhancers (e.g., liraglutide vs. semaglutide and pasireotide vs. octreotide).

To the best of our knowledge, this is the first article to highlight the discrepancy in the amount of absorption enhancers required to improve oral absorption between nonclinical animal experiments and clinical human studies where physical mixture solutions and solid formulations were administered, respectively. Although the dosage of SNAC that enhances the oral absorption of peptides is 100 to 1000 mg/kg in male rats, based on our experiments in which a combined solution of SNAC and peptides was orally administered to them, and the fact that absorption enhancers (SNAC-related compounds and C10) have generally been used at high oral doses ranging from 25 to 800 mg/kg in animals, such huge dosages are not feasible in humans. Only 300 mg of SNAC and around 100 mg of C8 (estimated amount) are contained in the Rybelsus tablets and Mycapssa capsules, respectively. The smaller amount of absorption enhancers required in humans compared with that in animals suggests that the current formulation technologies applied to humans are well designed to promote local disintegration and enhance absorption in the gastrointestinal tract. However, this also indicates that there is room for improvement. That is to say, new solid formulations that can maximize the oral absorption of middle-to-large molecules with the minimum required amount of SNAC and C8 or new absorption enhancers superior to SNAC and C8 should be developed to further promote the practical use of orally bioavailable middle-to-large molecules.

This review also covers the state-of-the-art molecular design and chemical modification approaches for oral delivery, since diversely structured drugs such as TPD and cyclic peptides have recently been developed as new modalities. Extensive research on the mechanism by which these molecules acquire membrane permeability is being performed, revealing the importance of their chameleonic property. Additionally, new evaluation methods (e.g., EPSA) have been developed to analyze these characteristics, along with new in silico criteria for an “extended” or “limit” of the rule of five. As illustrated by cyclosporin (oral BA: 30%), well-designed chemically modified middle-to-large molecules can achieve much higher oral BA than absorption enhancers (oral BA: typically ~1%). Thus, when considering the option of applying either an absorption enhancer or a chemical modification approach for middle-to-large molecules, the latter approach should be prioritized due to its potential for achieving higher oral BA. It has been asserted in the past that the hydrolytic metabolism of peptides by digestive enzymes has primarily been considered problematic, but our recent considerations additionally suggest that cyclic peptides undergo synergic elimination by P450 metabolism and P-gp efflux and middle-to-large molecules are readily recognized by P-gp. The latest knowledge on this class of molecules, such as cyclic peptides (e.g., LUNA-18 (MW: 1438), MK-0616 (MW: 1616), and PA-001 (MW: unknown)) and TPDs (e.g., ARV-110 (MW: 812) and ARV-471 (MW: 724)), should provide the key for the successful development of chemical modification approaches for their oral use.

In conclusion, we strongly believe that we will be able to fully utilize both absorption enhancer and chemical modification technologies to achieve the oral delivery of middle-to-large molecules. Since one of the most important aspects of drugs is their pharmacological activity, sometimes the molecular structures of new modalities cannot be drastically changed to maintain their affinity toward the target molecules (e.g., peptide hormones and oligonucleotides). In such cases, the utilization of absorption enhancer technology would be more effective for oral absorption than chemical modification. Meanwhile, if new modalities involving drastic chemical modification (e.g., cyclization and *N*-alkylation) can achieve sufficient pharmacological activity and obtain a chameleonic property to achieve membrane permeability, this type of molecule can be orally absorbed without absorption enhancers. It is highly anticipated that the optimal strategy will be employed for each new drug modality, taking into consideration its chemical structure and pharmacological activity to provide patients with orally bioavailable drugs with middle-to-large molecular sizes in the future.

## Figures and Tables

**Figure 1 pharmaceutics-16-00047-f001:**
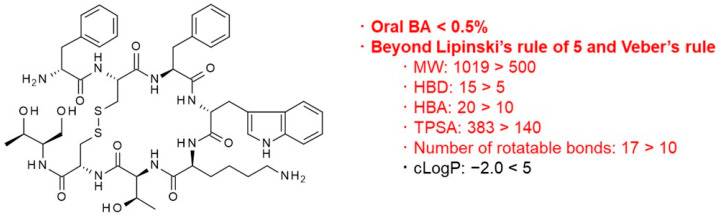
Chemical structure and physicochemical properties of octreotide. MW: molecular weight, HBD: number of hydrogen bond donors, HBA: number of hydrogen bond acceptors, TPSA: topological polar surface area, and cLogP: calculated octanol-water partition coefficient.

**Figure 2 pharmaceutics-16-00047-f002:**
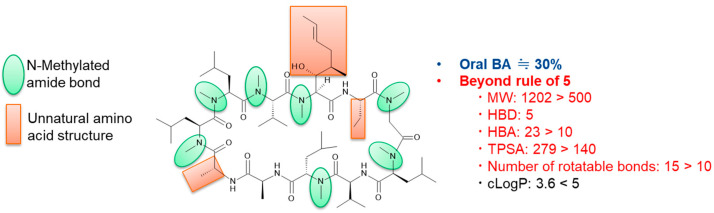
Chemical structure and physicochemical properties of cyclosporin. MW: molecular weight, HBD: number of hydrogen bond donors, HBA: number of hydrogen bond acceptors, TPSA: topological polar surface area, and cLogP: calculated octanol-water partition coefficient.

**Figure 3 pharmaceutics-16-00047-f003:**
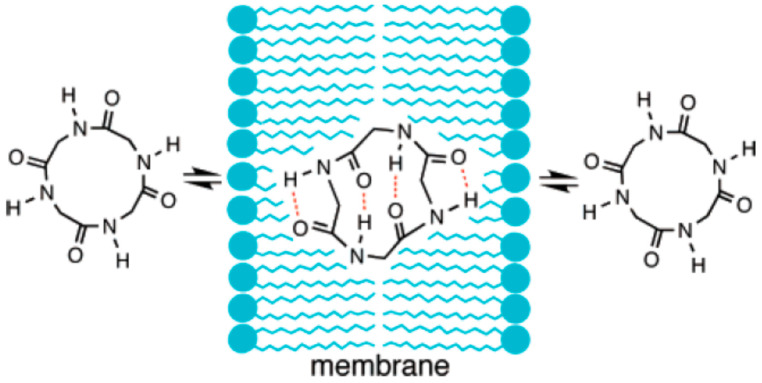
Schematic representation of the conformational basis of the membrane permeability of cyclic peptides (chameleonic property). Adapted with permission from [43]. Copyright (2006) American Chemical Society.

**Figure 4 pharmaceutics-16-00047-f004:**
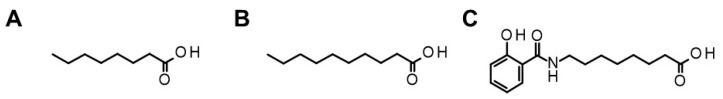
Chemical structures of C8 (**A**), C10 (**B**), and SNAC (**C**).

**Figure 5 pharmaceutics-16-00047-f005:**
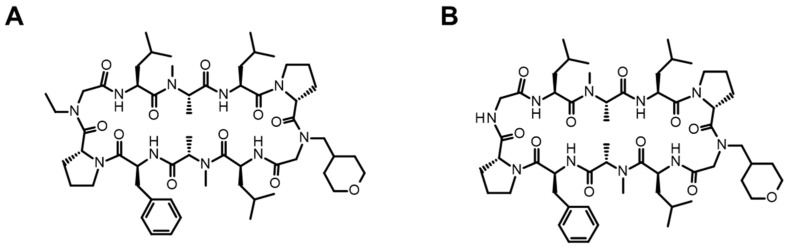
Chemical structures of cyclic peptide A ((**A**) MW: 1091) and its de-ethylated form ((**B**) MW: 1063). The above figure was cited from [121].

**Figure 6 pharmaceutics-16-00047-f006:**
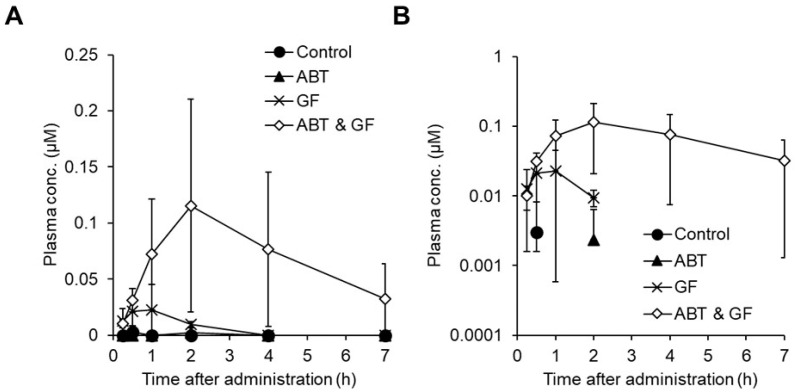
Pharmacokinetics of cyclic decapeptide A (MW: 1091) in mouse plasma after its oral administration at 1 mg/kg with or without ABT (P450 inhibitor) and/or GF (P-gp inhibitor). Plasma concentrations of cyclic decapeptide A were determined by LC-MS/MS and plotted. Each point represents the mean ± SD of three animals. (**A**) Normal plot; (**B**) semi-log plot. ABT and GF represent 1-aminobenzotriazole and GF120918, respectively. The above figure was cited from [121].

**Figure 7 pharmaceutics-16-00047-f007:**
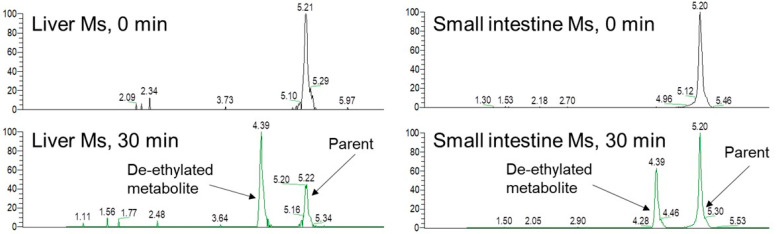
Metabolite identification of cyclic decapeptide A (MW: 1091) after incubation with hepatic and intestinal microsomes from mice. Ms represent microsomes. The above figure was cited from [121].

**Figure 8 pharmaceutics-16-00047-f008:**
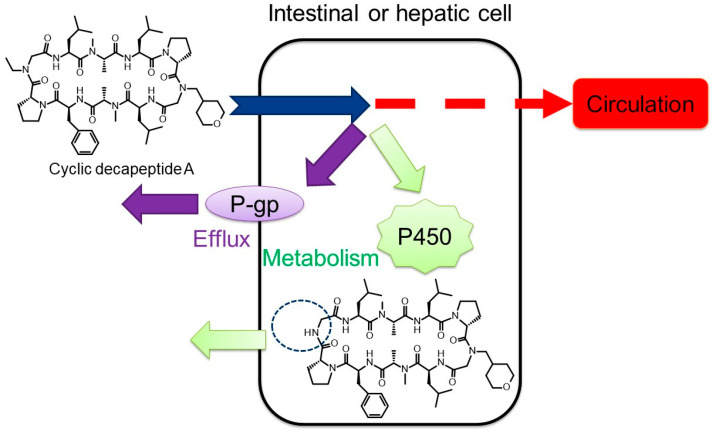
Synergic elimination of cyclic decapeptide A (MW: 1091) by P-gp and P450. The above figure was cited from [121].

**Figure 9 pharmaceutics-16-00047-f009:**
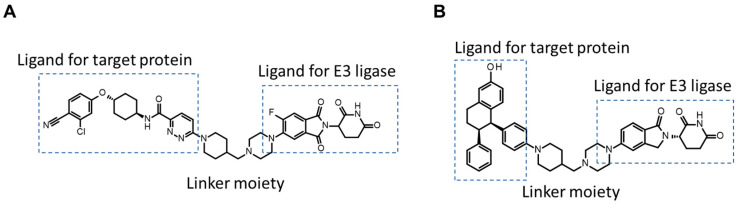
Chemical structures of TPDs ((**A**) ARV-110 (MW: 812); (**B**) ARV-471 (MW: 724)).

**Figure 10 pharmaceutics-16-00047-f010:**
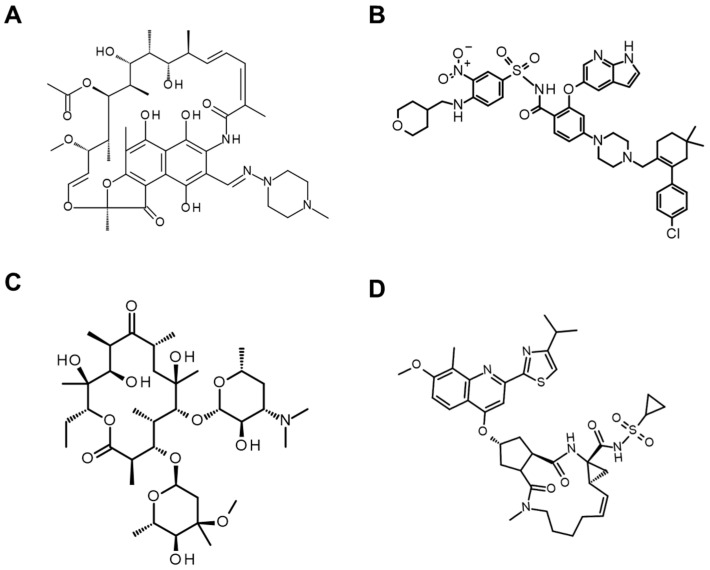
Chemical structures of other representative middle-to-large molecules. (**A**) Rifampicin (MW: 823), (**B**) venetoclax (MW: 868), (**C**) erythromycin (MW: 734), and (**D**) simeprevir (MW: 750).

**Figure 11 pharmaceutics-16-00047-f011:**
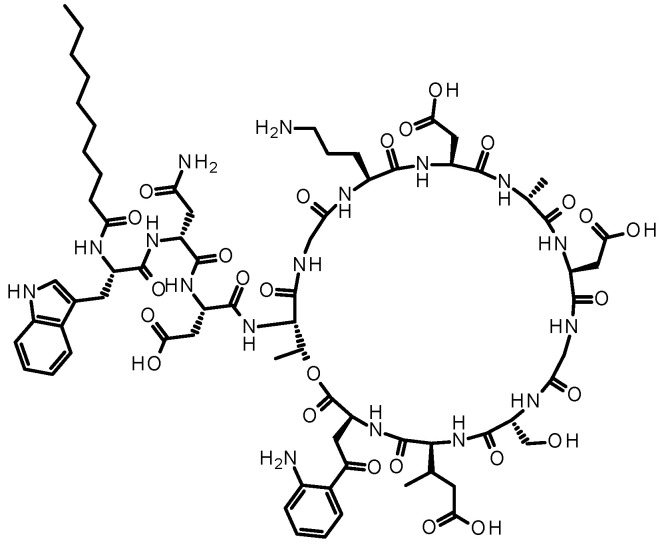
Chemical structure of daptomycin (MW: 1621).

**Figure 12 pharmaceutics-16-00047-f012:**
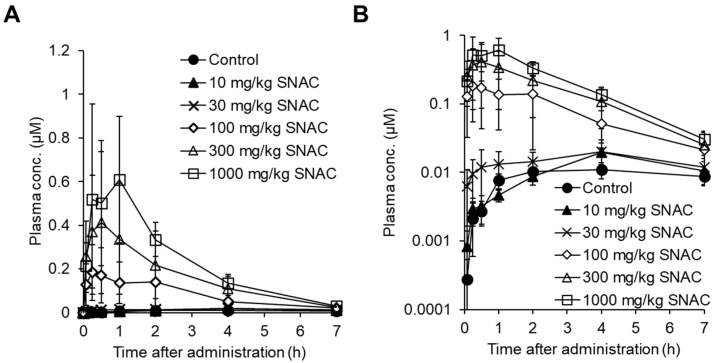
Pharmacokinetics of daptomycin (MW: 1621) in male rat plasma after its oral administration at 10 mg/kg with or without SNAC at doses ranging from 10 to 1000 mg/kg. Plasma concentration of daptomycin was determined by LC-MS/MS and plotted. Each point represents the mean ± SD of three animals. (**A**) Normal plot; (**B**) semi-log plot. The above figure was cited from [148].

**Figure 13 pharmaceutics-16-00047-f013:**
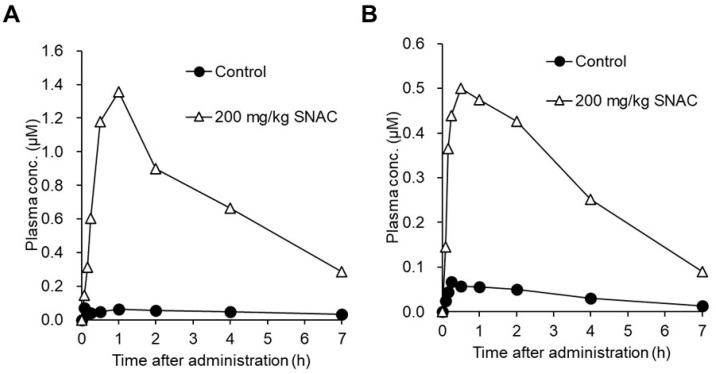
Time-dependent changes in the concentrations of daptomycin (MW: 1621) in monkey (**A**) and dog (**B**) plasma after its oral administration at 10 (**A**) and 5 (**B**) mg/kg with or without SNAC at 200 mg/kg. Plasma concentrations of daptomycin were determined by LC-MS/MS and plotted. Each point represents the mean of two animals. The above figure was cited from [121].

**Figure 14 pharmaceutics-16-00047-f014:**
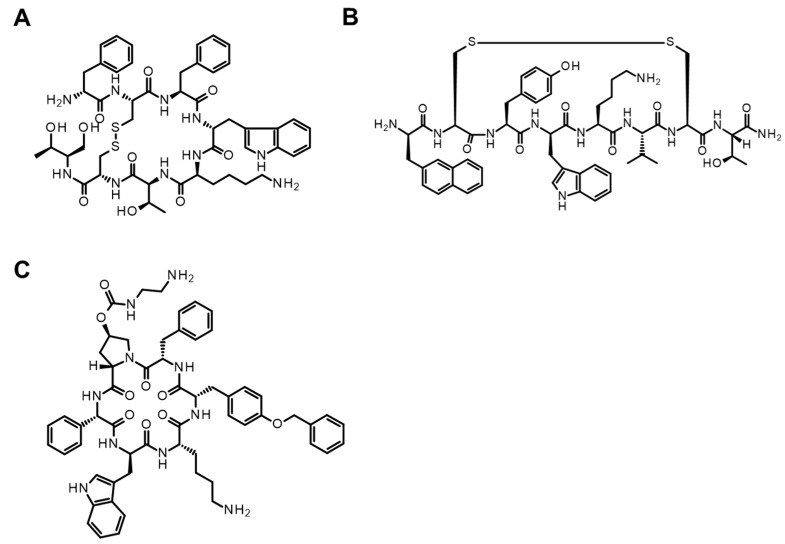
Chemical structures of octreotide ((**A**) MW: 1019), lanreotide ((**B**) MW: 1096), and pasireotide ((**C**) MW: 1047).

**Figure 15 pharmaceutics-16-00047-f015:**
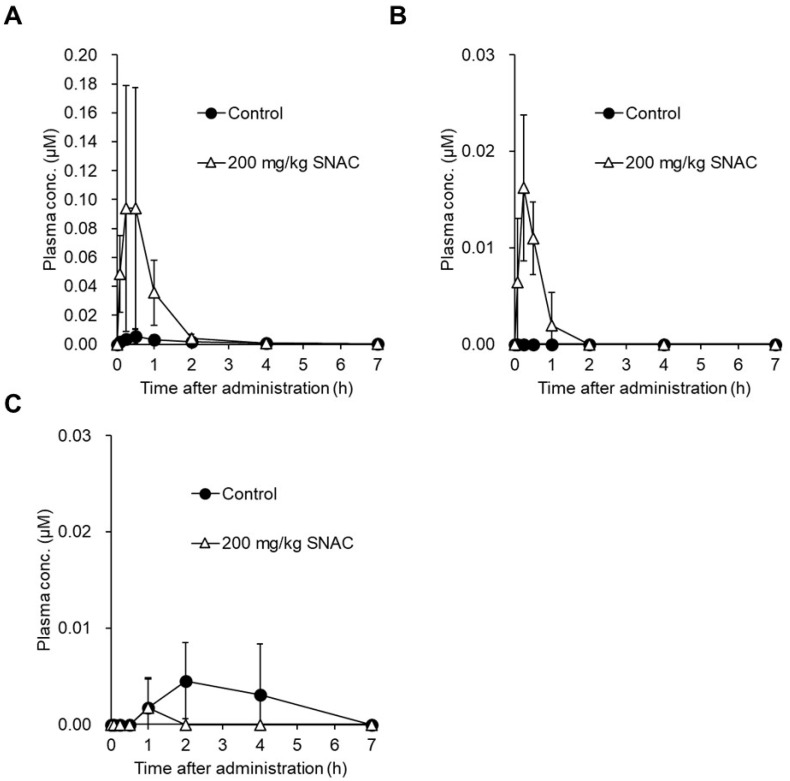
Time-dependent changes in the concentrations of octreotide ((**A**) MW: 1019), lanreotide ((**B**) MW: 1096), and pasireotide ((**C**) MW: 1047) in rat plasma after their oral administration at 5 mg/kg with or without SNAC at 200 mg/kg. Plasma concentrations of octreotide, lanreotide, and pasireotide were determined by LC-MS/MS and plotted. Each point represents the mean ± SD of three animals. The above figure was cited from [148].

**Figure 16 pharmaceutics-16-00047-f016:**
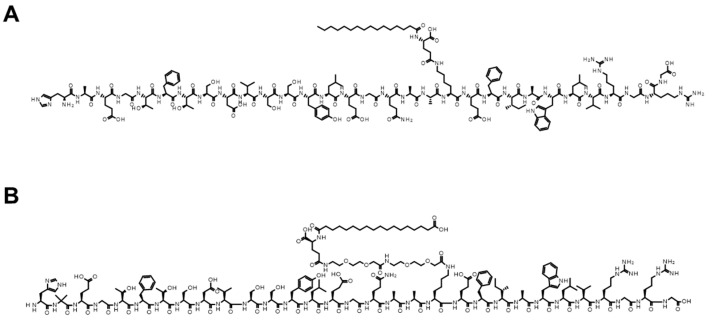
Chemical structures of liraglutide ((**A**) MW: ca. 3800) and semaglutide ((**B**) MW: ca. 4100).

**Table 1 pharmaceutics-16-00047-t001:** Advantages and limitations of chemical modifications and absorption enhancers for middle-to-large molecules.

	Advantages	Limitations
Chemical modifications	Oral bioavailability (BA) of chemically modified middle-to-large molecules tends to be higher than that by absorption enhancers. (In the best case, oral BA of cyclic peptide is 100% [36]).	Synthesis and structure design to acquire sufficient bioavailability are difficult because the following points need to be considered: ✓ Intramolecular hydrogen bonding and chameleonic property are sometimes necessary [37]. ✓ Molecular volume or weight should be less than 1500 Å^3^ and 1200 because of the limitations of solubility and permeability [38]. ✓ If the middle-to-large molecules possess poor solubility, special formulations such as self-microemulsifying drug delivery system (e.g., Neoral^®^) are necessary [39]. ✓ Avoidance of P-gp efflux and P450 metabolism is sometimes necessary (Section 3.1).
Absorption enhancers	Oral bioavailability of middle-to-large molecules whose molecular weight exceeds 1000 can be enhanced (e.g., molecular weight of semaglutide in Rybelsus tablets is more than 4100). Fatty acids (C8 and C10) and SNAC have acquired generally recognized as safe (GRAS) status from the FDA. Absorption-enhancing effect can be easily evaluated in animals (Section 3.2).	Oral bioavailability is relatively low, generally ranging from 1% to 20%. Interindividual variety in exposure [17] and high cost of goods (COG) [40] are sometimes problematic due to low bioavailability (Section 3.2). Development of sophisticated formulations utilizing absorption enhancers is challenging. Patented formulations (e.g., Eligen^®^ and TPE^TM^) might be necessary (Section 2.3). Safety of some absorption enhancers is unknown. Detailed mechanism and structure–activity relationship for absorption enhancement are unknown (Section 3.2).

**Table 2 pharmaceutics-16-00047-t002:** Analytical method to evaluate chameleonic property.

Evaluation Method	Detail	Reference
Partition coefficient in octanol-water (LogP_oct_) and toluene/water (LogP_tol_)	The difference between LogP_oct_ and LogP_tol_ (∆LogP) correlates with the presence or absence of intramolecular hydrogen bonding.	[49]
Molecular (3D) polar surface area in nonpolar environments (MPSA) and topological polar surface area (TPSA)	TPSA is a polar surface area calculated as a sum of fragment-based contributions. MPSA is the minimal solvent-accessible polar surface area in 3D conformations. If the value of TPSA minus MPSA (∆PSA) is larger than 0.2 × molecular weight—140 Å^2^ or TPSA—140 Å^2^, the evaluated middle-to-large molecules would possess chameleonic property.	[37]
Lipophilic permeability efficiency (LPE)	LPE is an index of the membrane permeability of middle-to-large molecules. It can be calculated as follows: LPE = distribution coefficient in decadiene-water at pH 7.4 − m_lipo_ (scaling factor) × calculated LogP_oct_ + b_scaffold_ (scaling factor).	[44]
Experimental polar surface area (EPSA)	EPSA is an index of the membrane permeability with consideration of intramolecular hydrogen bonding. It can be measured by supercritical fluid chromatography.	[45,46,50,51]
Nuclear magnetic resonance (NMR) analysis	Amide temperature coefficients and H/D exchange study measured by NMR indicate the presence or absence of intramolecular hydrogen bonding.	[47,48]
X-ray analysis	Three-dimensional structure of middle-to-large molecules can be elucidated by X-ray crystallography, indicating the presence or absence of intramolecular hydrogen bonding.	[36]
In silico structural simulation	Molecular dynamic method can predict chameleonic property or membrane permeability.	[52,53]

**Table 3 pharmaceutics-16-00047-t003:** Representative absorption enhancers.

Absorption Enhancer	Mechanism	Available Safety Information
SNAC and related compounds (4-CNAB and 5-CNAC)	Enhancing transcellular permeation [61,62,63]	NOAEL of SNAC: 500 and 500 mg/kg/day in male and female mice, 500 and 75 mg/kg/day in male and female rats, and 300 and 300 mg/kg/day in male and female monkeys, respectively [64].
C8, C10, fatty acids, and surfactants	Opening tight junctions and/or causing membrane perturbation [65,66]	LD_50_ of C8 and C10: 1280–10,080 mg/kg [67] and 3730 mg/kg [68] in rats, respectively.
Amino acids (arginine and tryptophan)	Unclear (possible involvement of receptor- or transporter-mediated uptake) [69,70]	NOAEL of arginine and LD_50_ of tryptophan: 3131 mg/kg in rats [71] and 5000 mg/kg in mice [72], respectively.
Acylcarnitines, EDTA, bile acid, NO, chitosan (polysaccharide), claudin modulator, 1-phenylpiperazine	Opening tight junctions [55,56,57,73,74,75,76]	LD_50_ of carnitine, EDTA, deoxycholic acid, nitroprusside, chitosan, claudin modulator, and 1-phenylpiperazine: 19.2 g/kg in mice [77], 2 g/kg [78], 1 g/kg in mice and rats [79], 43 mg/kg in mice [80], 16 g/kg in mice [81], unknown, and 210 mg/kg in rats [82], respectively.
TAT, octa-arginine, and related peptides (cell-penetrating peptides: CPPs)	Inducing macropinocytosis [54,60]	Unknown
Cyclic DNP peptide (CPPs)	Inducing macropinocytosis [58,59]	Unknown
Intravail^®^ (alkylsaccharide excipient)	Opening tight junctions and enhancing transcellular permeation [83,84]	LD_50_ of Intravail^®^: 2000 mg/kg in rats [85].
Citric acid and protease inhibitors	Protecting peptides and proteins from digestive enzymes [86]	LD_50_ of citric acid: 5040 and 3000 mg/kg in mice and rats, respectively [87].

**Table 4 pharmaceutics-16-00047-t004:** Representative formulations for absorption enhancement.

Formulation	Composition and Design	API	Marketed	Reference
Rybelsus^®^ and Eligen^®^ B12	Immediate-release tablet with SNAC	Semaglutide (MW: ca. 4100) and vitamin B12 (MW: 1355)	Yes	[17,18,20,21,35,108,110]
enTRinsic™	Enteric-coated capsule composed of cellulose acetate phthalate	Esomeprazole (MW: 345)	No	[20,111]
GIPET™	Enteric-coated tablet with various additives (C10, etc.)	Heparin (MW: ca. 1000–35,000), I338 (MW: ca. 6400), acyline (MW: ca. 1500), and GLP-1 (MW: ca. 3000–4000)	No	[20,24,112]
POD™ (Protein Oral Delivery)	Enteric-coated capsule with various additives (SNAC, EDTA, aprotinin, fatty acid, trypsin inhibitor, etc.)	Insulin (MW: ca. 5800) and exenatide (MW: ca. 4200)	No	[20,113,114]
Peptelligence™ and Ovarest^®^	Enteric-coated tablet with various additives (acylcarnitine, citric acid, etc.)	Salmon calcitonin (MW: ca. 3400), leuprolide (MW: ca. 1200), and difelikefalin (MW: ca. 680)	No	[20,21]
TPE™ and Mycapssa^®^	Enteric-coated capsule containing oily suspension of C8 and additives	Octreotide (MW: ca. 1000)	Yes	[19,20,21,35]
Nodlin^TM^	Enteric-coated nanoparticle	Insulin (MW: ca. 5800)	No	[26]
Capsulin™	Enteric-coated capsule with bile salt and antioxidant	Insulin (MW: ca. 5800)	No	[35,115]
SmPill^®^	Emulsion-based formulation containing various absorption enhancers (sodium taurodeoxycholate, C10, etc.)	Salmon calcitonin (MW: ca. 3400) and cyclosporin (MW: 1202)	No	[116,117]
Oraldel™	Cyanocobalamin-coated nanoparticle consisting of carbohydrate-based sugar	Insulin (MW: ca. 5800)	No	[35]
HDV (hepatocyte-directed vesicle) and other liposomes	Liposome composed of hepatocyte-targeting molecule (disofenin, etc.), various phospholipids and/or cholesterol	Insulin (MW: ca. 5800)	No	[35,118,119]

API represents active pharmaceutical ingredient.

## Data Availability

Not applicable.

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
