# Peer review of "Oral Absorption of Middle-to-Large Molecules and Its Improvement, with a Focus on New Modality Drugs"

_pharmaceutics, 2023, doi:10.3390/pharmaceutics16010047_

Round 1
Reviewer 1 Report
Comments and Suggestions for Authors
This is a nice review of the field. Please consider the following when revising the manuscript:
1. Include the amount of the sodium caprylate in Mycapssa. This information could be dug out from their patents.
2. I suggest the authors to change macromolecules to large molecules. The former usually refers to proteins and polysaccharides.
3. Cyclosporin products have a good bioavailability due to its unique structure AND novel formulations.
4. Include APIs in Table 4.
5. TPDs (Section 3.1.2) are small molecules. They should not be included.
6. Not sure you can call those in Figure 10 macromolecules.
7. Whenever possible, please add molecular weight information, such as those in Figures, Tables, and text. (including those for oligonucleotides).
Author Response
Thank you for your comments.
Answers to your comments are described in a word file.

Reviewer 2 Report
Comments and Suggestions for Authors
In the current manuscript, the authors describe the use of absorption enhancers for improved oral delivery of macromolecule drugs. They summarized previously published reports on the oral delivery of small and macromolecules, concluding with the observation that most oral formulations do not progress to the clinic due to higher doses and the cost of formulation development.
I would suggest including some insight into the toxicity associated with the described absorption enhancers.
Author Response
Thank you for your comment.
Answer to your comment is described in a word file.

Reviewer 3 Report
Comments and Suggestions for Authors
This manuscript provide a commentary on macromolecules.
I am unconvinced what this manuscript brings to the field as much of the work is already published in greater detail. I realise that this is a review but I cannot see the focus of this review.
It says that the approaches and recent advances of orally bioavailable macromolecular drugs will be reviewed. I think this review could be improved by the inclusion of a checklist of why certain studies are included/excluded as at present it is not comprehensive yet there is no logic on what is or is not presented.
Perhaps if it was limited to only regulatory approved drugs with a deep dive into the mechanisms of how their absorption is improved that would provide focus - or to expand the remit to studies on agents in Phase 3 studies and do the same deep dive would be useful.
Ultimately it is up to the editors to decide if the readership would like this commentary
In terms of specific revisions that I would suggest:
Table 3 include the drugs where these absorption enhancers have been used and the way in which they were evaluated (eg human study; animal study or other study)
Fatty acids C8 and C10 line 154 - be explicit and state which products or which drugs including a clinical trial so that the reader gets the level of detail required
Table 4 include details of the drugs that are contained in each formulation and whether these are now commercially available or where they are in the pipeline of testing
Section 3 line 239. Be explicit on how you define successful cases - does this mean regulatory approval or success in clinical trials. It is not clear to the reader how this was defined or how the relevant literature to include was identified. Perhaps include a link to your search strategy?
In the conclusions section it is just a repeat of the introduction as the paper has not added value so nothing can be concluded from your own work. It woudl be useful to say what the status is based on your literature screening and to define some current parameters.
Specifically in the conclusion you state that there are high manufacturing costs but that was only very briefly mentioned in your review so I am not sure it can be part of your conclusion
Author Response

(The authors gave the same response as above.)
